

# A novel signaling transduction pathway of melatonin on lactose synthesis in cows via melatonin receptor 1 (MT1) and prolactin receptor (PRLR)

Yunjie Liu[1], Songyang Yao[1], Qinggeng Meng[2], Xuening Liu[1], Huigang Han[1], Chunli Kan[3], Tiankun Wang[4], Wenjuan Wei[5], Shujing Li[6], Wenli Yu[6], Zengyuan Zhao[6], Changwang He[5] and Guoshi Liu[1]

[1] China Agricultural University, Beijing, China
[2] Beijing Changping District Nankou Town Agricultural Clothing Center, Beijing, China
[3] Beijing Changping District Shisanling Town Agricultural Clothing Center, Beijing, China
[4] Beijing Changping District Animal Disease Prevention and Control Center, Beijing, China
[5] Beijing Sunlon Livestock Development Co. Ltd., Beijing, China
[6] Shijiazhuang Tianquan Elite Dairy Co. Ltd., Shijiazhuang, China

Corresponding author
Guoshi Liu, gshliu@cau.edu.cn

## ABSTRACT

In the current study, we explored the relationship between melatonin and lactose synthesis in *in vivo* and *in vitro* conditions. We found that long-term melatonin feeding to the dairy cows significantly reduced the milk lactose content in a dose dependent manner. This lactose reduction was not associated with a negative energy balance, since melatonin treatment did not alter the fat, glucose, or protein metabolisms of the cows. To identify the potential molecular mechanisms, the cow's mammary epithelial cells were cultured for gene expression analysis. The results showed that the effect of melatonin on lactose reduction was mediated by its receptor MT1. MT1 activation downregulated the mRNA expression of the prolactin receptor gene (PRLR), which then suppressed the gene expression of SLC35B1. SLC35B1 is a galactose transporter and is responsible for the transportation of galactose to Golgi apparatus for lactose synthesis. Its suppression reduced the lactose synthesis and the milk lactose content. The discovery of this signal transduction pathway of melatonin on lactose synthesis provides a novel aspect of melatonin's effect on carbohydrate metabolism in cows and maybe also in other mammals, including humans.

# INTRODUCTION

Lactose is a primary carbohydrate in milk. Its concentration is directly associated with milk production by maintaining the osmotic pressure of milk (*Zhao & Keating, 2007*; *Shahbazkia et al., 2010*). The content of milk lactose can also serve as a health biomarker and reproductive capacity of cows (*Televičius et al., 2021*). For example, the level of milk lactose indicates subclinical and clinical ketosis (*Steen, Osteras & Gronstol, 1996*) and provides the most informative feature for estimating energy balance (*Heuer et al., 2000*;

*Reist et al., 2002*). In addition, the relationship between milk lactose content and fertility of the cows has been studied with the observation that the lactose ratio in milk is positively correlated with pregnancy rates of the cows (*Buckley et al., 2003*).

In addition, milk lactose is an indicator of a cow's health. For example, mastitis is a common disorder in dairy cows, and this disorder can significantly impact milk quality with its altered lactose content (*Antanaitis et al., 2021*). Mastitis is usually treated with antibiotics. The residuals of the antibiotics in milk are serious biohazards for human health. The current food safety policy requires reducing the residuals of antibiotics in milk. As a result, alternative ways to treat mastitis are required. One of these alternative methods to treat mastitis is melatonin. Melatonin is a well-known free radical scavenger and antioxidant (*Royano & Reiter, 2010*). It also acts as an immunoregulator in mammals to exert immunostimulatory or immunosuppressive effects depending on the conditions (*Tan et al., 2016*; *Luchetti et al., 2010*; *Yang et al., 2013*; *Tajes et al., 2009*). It has been used to alleviate mastitis in dairy cows. Melatonin as an adjunctive treatment agent has shortened the duration of treatment (*Yang et al., 2017*). Both subcutaneous melatonin injections (*Wu et al., 2021*) and rumen bypass melatonin feeding (*Yao et al., 2020*) significantly reduced somatic cell counts in milk and high somatic cell counts in milk are an indicator of mastitis. The anti-inflammatory effect of melatonin on mastitis also impacts the lactose content of milk.

It has been reported that continuously feeding rumen bypass melatonin to the cows not only significantly reduced mastitis but also the lactose content in milk (*Yao et al., 2020*). Auldist et al. reported that subcutaneous implantation of melatonin in cows lowered their milk lactose (*Auldist et al., 2007*) and similar results were observed in dairy goats and sheep (*Molik, Błasiak & Pustkowiak, 2020*; *Yang et al., 2020*). Although melatonin supplementation with reduced milk lactose content in cows or other species has been frequently reported, the underlining mechanism(s) are currently unknown. As glucose metabolism is involved in the energy balance of the cow, we hypothesize that melatonin reduces lactose in milk, possibly by affecting the energy balance of the cow or simply by affecting the synthesis of lactose in the mammary gland.

In this study, we systemically investigated the association between melatonin levels and milk lactose in cows. These include alterations of milk lactose with time, the cow's reproductive activity, and energy balance indexes affected by different doses of melatonin feeding. In addition, an *in vitro* lactose synthetic model will be established to investigate the molecular mechanisms related to melatonin and lactose production. We believe that the results will provide further insight into the effects of melatonin on lactose synthesis in milk.

## MATERIALS & METHODS

### Chemicals and agents

Melatonin, luzindole and 4P-PDOT were purchased from Sigma-Aldrich (St. Louis, MO, USA) and then dissolved in dimethyl sulfoxide (DMSO), respectively. DMEM/F12 and FBS were purchased from Gibco. Prolactin and insulin were purchased from CUSABIO (Wuhan, China).

## Animals

The research was performed by the provisions of the China Agricultural University Laboratory Animal Welfare and Animal Experimental Ethical Inspection Committee. The study approval number is AW01502202-1-1.

The study was conducted at the Beijing Sunlon Livestock Development Company, China. Twenty healthy Holstein cows with similar body conditions, second parity, $584 \pm 24$ kg of weight, and $50 \pm 20$ days of lactation were selected. The cows were exposed to the natural light/dark cycles without human interference and were fed the TMR diet four times a day at 6:00, 12:00, 18:00, 22:00, and milked three times a day at 7:30, 14:30, 21:30, accessed to water ad libitum. The composition and nutrient levels of the diet are shown in Table 1. The nutritional requirements of dairy cows met the standard in *NRC (2001)*. Referring to the random numbers table, they were randomly divided into four groups 5 for a group. Three experimental groups were given 2.5, 5, and 10 mg/kg/d of oral melatonin/day respectively, at 9:00 a.m. for 28 days. After that, the melatonin feeding was stopped for 21 days. At the end of the experiment, all experimental cows returned to normal production.

Dairy cows were fed melatonin individually to ensure the dose was accurate. Before feeding melatonin, the prepared melatonin powders were quantitatively loaded into a paper package (digestible) according to the experimental group and the body weight of each cow. The cow is secured with a neck clip, standing on her side with one hand opening her mouth and the other hand placing the paper packet containing melatonin into her pharynx so that she swallows it completely. During the experiments, no adverse reactions were observed in any of the animals. Control cows were fed digestible paper packets without melatonin. Milk samples and blood samples were collected every 7 days prior to melatonin feeding at about 7:30 a.m. until the termination of the study.

## DHI Measure

DHI determination was performed by the National Milk Product Standard Sanction Laboratory located at the Beijing Animal Husbandry Station using a DHI measuring instrument (MilkoscanFT1, Serial No.91755049, Part No.60027086, made in Denmark). The somatic cell count in milk was determined using the Fossomatic™ FC (Serial No.91755377, Part No.60002326, made in Denmark) automated tester (*Wu et al., 2021*).

## Serum biochemistry and hormone test

After fresh blood collection, the serum was obtained by centrifugation at 3500 r/min for 5 min and stored at $-20\,°C$. The stored samples were melted at room temperature and tested with the methods described below. Serum melatonin was measured with Agilent 6470 liquid chromatography-tandem MS (LC-MS-MS) (Agilent Technologies, Santa Clara, CA, USA). Non-esterified fatty acid (NEFA) was measured by NEFA kit (Randox Laboratories Ltd, Crumlin, Co. Antrim, UK), following the manufacturer's instructions, the growth hormone was measured with a commercial ELISA (MBS703041; MyBiosource, San Diego CA, USA). Plasma was analyzed for glucose (catalog no. 439–90901; Wako Chemicals USA, Richmond, VA, USA) and prolactin by enzyme-linked immunosorbent assay (Bovine Prolactin ELISA; MBS2022462; MyBioSource.com). Progesterone was measured by radioimmune assay

**Table 1 Ingredient composition and nutrient content of the basal diets (DM basis).**

| Items | Value (%) |
|---|---|
| Ingredients | |
| High yield concentrate[a] | 39.1 |
| Whole corn silage | 26.8 |
| Alfalfa hay | 12.5 |
| Steamflaked corn | 11.6 |
| Fat power | 2.1 |
| Oat grass | 1.6 |
| Extruded soybean meal | 1.6 |
| Molasses cane | 1.6 |
| Soybean hull | 1.2 |
| Beet pulp | 1.1 |
| Sunflower meal | 0.8 |
| Total | 100 |
| Nutrient levels | |
| NDF | 30.7 |
| ADF | 20.4 |
| CP | 17.8 |
| EE | 5.9 |
| Starch | 27.3 |
| Calcium | 1.0 |
| Phosphorus | 0.5 |

**Notes.**

[a]High yield concentrate was provided by Beijing Capital Agribusiness Group, including (DM basis): CP 24.75%, NDF 21.11%, Starch 36.26%, Fat 3.41%, Ash 7.54%, Calcium 1.04%, Phosphorus 0.53%, NaCl 1.0%, Fe 105 mg/kg, Zn 65 mg/kg, Mn 24 mg/kg, Cu 7 mg/kg, Mg 2 g/kg, K 10 g/kg, VA 20,000 IU/kg, VD 2300 IU/kg, and VE 88 IU/kg.

(RIA; KIP1458; Diasource, Nivelles, Belgium), and insulin was determined by a 125I-insulin RIA CT kit (CIS Bio International Ltd, Gif-Sur-Yvette, France). All measurements followed the manufacturer's instructions.

## Cell culture and treatments

The MAC-T cell line was a gift from Prof. Ying Yu (China Agriculture University, Beijing, China). MAC-T cell were cultivated in complete medium, containing DMEM/F12 (12500062; Gibco, Billings, MO, USA), 10% FBS, 1 $\mu$g/mL hydrocortisone, 5 $\mu$g/mL insulin, 100 $\mu$g/mL streptomycin, 100 $\mu$g/mL penicillin and streptomycin (*Johnson et al., 2010*). Cells were incubated under a humidified atmosphere of 95% air and 5% $CO_2$ at 37 °C for subsequent experiments. The complete medium was then replaced with lactogenic medium (complete medium supplemented with 5 $\mu$g/mL prolactin) 48 h before cells were treated. To identify whether melatonin and its receptors were involved in lactose synthesis, MAC-T cell were treated for 24 h with 10 $\mu$mol/L melatonin and luzindole (nonselective melatonin membrane receptor antagonist) or 4P-PDOT (MT2-selective melatonin receptor antagonist), respectively (*Wang et al., 2019*).

**Table 2 Primer sequences.**

| Gene name | F | R |
|---|---|---|
| LALBA | AGACTTGAAGGGCTACGGA | TAGTTGCTTGAGTGAGGGTT |
| B4GALT1 | TGCCCTGAGGAGTCCCC | GGCCACCTTGTGAGGAGAG |
| SLC35A2 | GTGGTCCAGAATGCTTCCCTC | CCAGGTGCTTCACGTTACCC |
| SLC35B1 | GACCTGCTCCATCATCACCAC | AGACCGAGACCCAAGAACACC |
| SLC2A1 | GACACTTGCCTTCTTTGCCA | AACCTAATGGAGCCTGACCC |
| PRLR | CCAAGCTCGTTAAATGTCGGT | AGGAGGCTCTGGTTCAACTATGT |

## Total RNA extraction and real-time PCR

Total RNA in MAC-T cell was extracted with TRIzol reagent (Invitrogen, Waltham, MA, USA). First-strand cDNA was synthesized with the reverse transcription kit (TaKaRa, Tokyo, Japan). Primers were designed using Primer Premier 5.0(PREMIER Biosoft, Palo Alto, CA). The primer sequence is shown in (Table 2). Quantitative real-time PCR was performed in the Real-Time Thermal Cycler CFX96 Optics Module (Bio-Rad Laboratories Inc., Hercules, CA, USA). Reactions were performed using the following conditions: 60 s of predenaturalization at 95 °C, followed by 40 cycles of 10 s denaturation at 95 °C and 30 s annealing and extension at 60 °C. Relative expression values were obtained using the average of three reference genes and the $2^{-\Delta\Delta CT}$ method. All of the above operations were performed according to the protocols issued by the manufacturers.

## Statistical analysis

All data are presented as mean ± SEM. Statistical analyses were performed using SPSS 25.0 (IBM SPSS Statistics, Armonk, NY, USA). Before the statistical analysis, the normal distribution and homogeneity of variables were checked. If the data were divided into two groups, Student's $t$-test was used. Otherwise, after passing the test for equal variances, statistical analysis was performed with one-way ANOVA with Tukey's post-hoc tests. Difference was considered significant at $P < 0.05$, highly significant at $P < 0.01$. The graphs were plotted using GraphPad Prism 7.0 software (GraphPad Software Inc., La Jolla, CA).

# RESULTS

## Effect of melatonin feeding on milk composition of cows

The results showed that the effects of melatonin on the composition of milk were dose-dependent. Melatonin feeding at doses of 2.5 mg/kg/d and 5 mg/kg/d significantly lowered milk lactose content and increased milk fat content compared to the control cows ($p < 0.05$, $n = 5$) (Figs. 1A and 1C). It appeared that the effects of melatonin at 2.5 mg/kg/d were more profound than that those at 5 mg/kg/d. The lactose content started to decrease at 14 days and maximized at 28 days after melatonin feeding, with a 13.5% reduction compared to the control ($p = 0.003$, $n = 5$). Surprisingly, the significant lower lactose content was still observed 7 days after termination of melatonin feeding ($p = 0.024$, $n = 5$). Melatonin feeding at the dose of 10 mg/kg/d had no significant effects on the milk lactose and fat

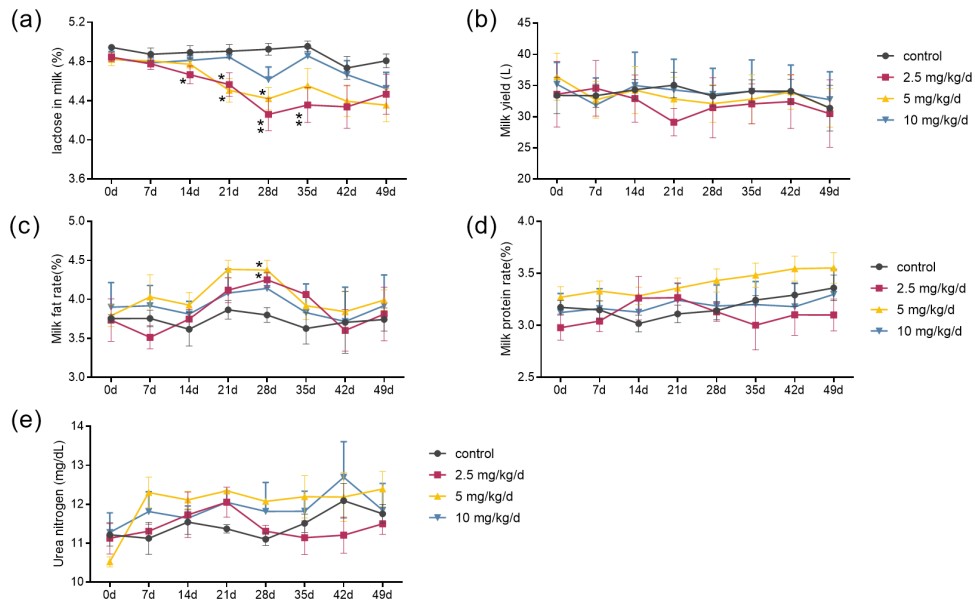

**Figure 1  Effect of melatonin treatment on milk composition.** (A–E) Milk lactose, daily milk yield, milk fat content, milk protein changes, and urea nitrogen content, respectively. ($N = 5$), *<0.05, **<0.01 *vs* control group.

content, and melatonin feeding at any dose had no significant effect on milk yield, milk protein, or urea nitrogen compared to the controls (Figs. 1B, 1D and 1E).

## Effect of melatonin feeding on serum nonestesterified fatty acid (NEFA) and glucose (GLU) in cows

Serum melatonin levels increased after melatonin feeding in a dose-dependent manner, and the significant difference compared to the control was only observed at the dose of 10 mg/kg/d group ($p < 0.05$, $n = 5$, Fig. 2A). Melatonin feeding did not significantly affect milk melatonin levels (Fig. 2B). Melatonin feeding also had no significant effects on serum NEFA, GLU levels compared to the control group (Figs. 2C–2F).

## Effect of melatonin feeding on serum hormones related to lactose synthesis

We examined the serum hormones that affect lactose synthesis, including prolactin, progesterone, insulin, and growth hormone. Prolactin levels decreased in all groups fed with melatonin compared to controls. The most significant decrease was observed in the melatonin 2.5 mg/kg/d group compared to the control group ($p = 0.036$, $n = 5$), followed by the 5 mg/kg/d group (Figs. 3A and 3C). Notably, serum progesterone levels significantly increased in the 2.5 mg/kg/d melatonin feeding group compared to the control group ($p = 0.016$, $n = 5$) but this increase had not been observed in both the 5 and 10 mg/kg/d melatonin feeding groups (Figs. 3B and 3D). Melatonin feeding had no significant effects on serum insulin and growth hormone levels compared to the control group (Figs. 3E–3H).

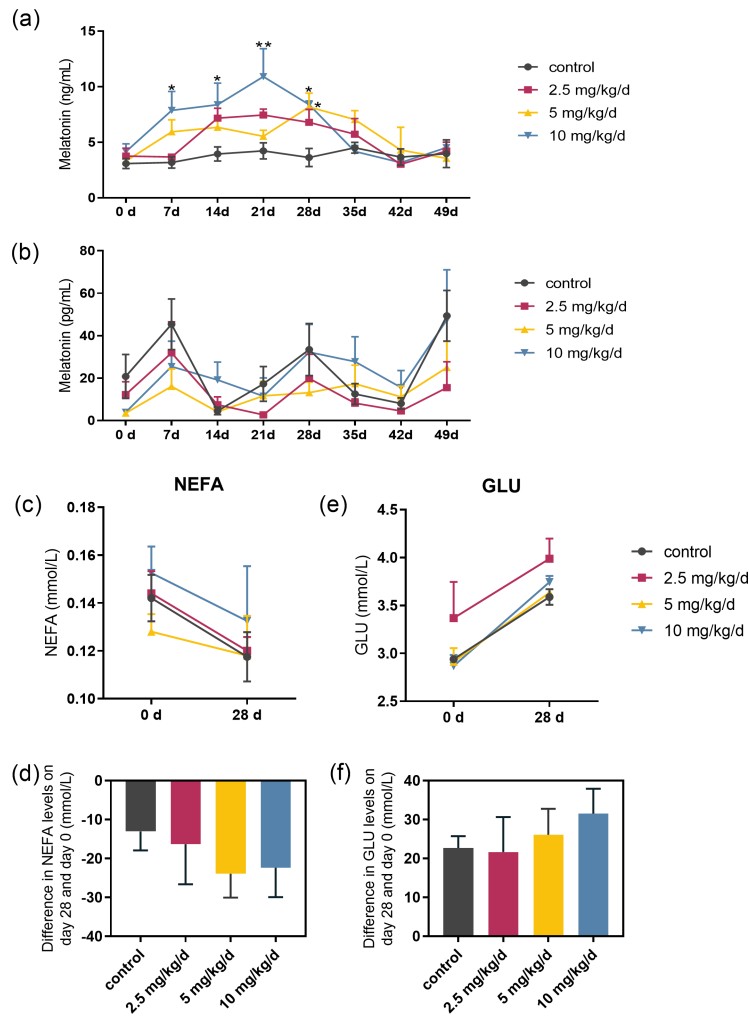

**Figure 2 Effects of melatonin feeding on serum NEFA and GLU levels in cows.** (A) Serum melatonin levels; (B) milk melatonin levels; (C, D) serum NEFA and GLU levels at 0 and after 28 days of melatonin feed-ing; (E, F) difference in NEFA and GLU levels in serum on day 28 and day 0. Each column represents the difference between the NEFA or GLU value of each animal tested on day 28 and the value tested on day 0. ($N = 5$), * <0.05, ** <0.01 *vs* control group.

## Effect of melatonin on lactose synthesis and its related genes expression in mammary epithelial cells

The effects of melatonin on lactose synthesis in dairy cow mammary epithelial cells were investigated. The lactose level in the medium of cultured cow mammary epithelial cells with melatonin treatment was significantly lower than that in the control group ($p= 0.011$, $n= 3$, Fig. 4A). In the lactating mammary gland, lactose is synthesized from glucose, and glucose transporter 1 (GLUT1) is essential for glucose uptake in epithelial cells. Melatonin treatment did not significantly affect GLUT1 compared to the control group. However, prolactin receptor expression in melatonin treated cells decreased by 35.5% compared to the control cells ($p = 0.009$, $n= 3$, Fig. 4B). Uridine diphosphate-galactose is actively transported into

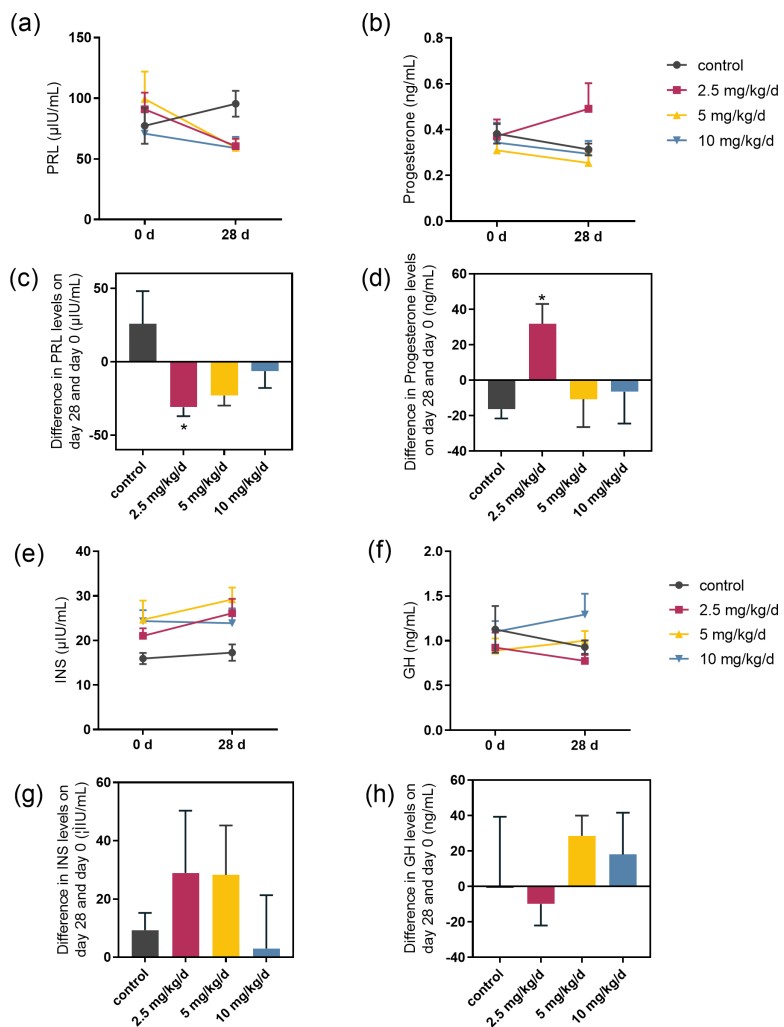

**Figure 3** **Effect of melatonin feeding on serum hormone levels.** (A, B) PRL and progesterone levels at 0 d and 28 d after melatonin feeding; (C, D) Difference in PRL and progesterone levels in se-rum on day 28 and day 0. Each column represents the difference between the PRL or progesterone value of each animal tested on day 28 and the value tested on day 0; (E, F) Insulin and growth hormone level at 0 and 28 d after melatonin feeding; (G, H) Differ-ence in insulin and growth hormone levels in serum on day 28 and day 0. Each column represents the difference between the insu-lin or growth hormone value of each group tested on day 28 and the value tested on day 0. ($N = 5$), * $<0.05$ *vs* control group.

the Golgi lumen by solute carrier family 35 member A2(SLC35A2) and solute carrier family 35 member B1 (SLC35B1) (*Lin et al., 2016*). SLC35B1 mRNA levels significantly decreased after melatonin treatment ($p = 0.011$, $n = 3$), whereas SLC35A2 mRNA levels did not significantly change compared to the control. In the Golgi compartment, lactose synthesis is catalyzed by lactose synthase, a complex of $\beta$-1, 4-galactosyltransferase($\beta$4GalT-I), and the essential cofactor $\alpha$-lactalbumin ($\alpha$-LA) (*Shahbazkia, Aminlari & Cravador, 2012*). Lactalbumin alpha (LALBA) mRNA level was upregulated by 269.7% in mammary epithelial cells with melatonin treatment compared to the control ($p < 0.001$, $n = 3$),

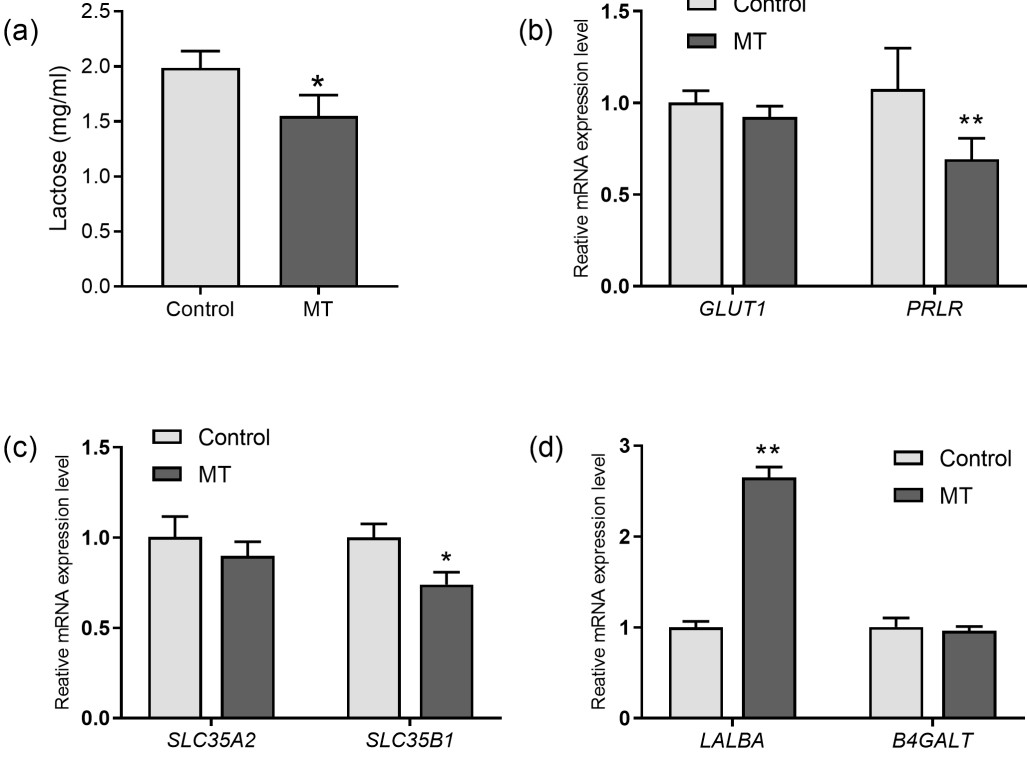

**Figure 4 Effect of melatonin on lactose synthesis and its related genes (mRNA) expression in mammary epithelial cells.** (A) Lactose level; (B) gene expression levels of GLUT1 and PRLR; (C) expression levels of galactose transporter genes SLC35A2 and SLC35B1; (D) expression levels of genes encoding lactose synthase LALBA and B4GALT, data presented as mean ± SEM. ($N = 3$), * <0.05, ** <0.01 *vs* control group.

whereas expression of *B4GALT* remained relatively unchanged with melatonin treatment (Fig. 4D).

## Effect of melatonin receptors on PRLR, LALBA and SLC35B1

Next, we evaluate whether the melatonin receptors MTNR1A (MT1) and MTNR1B (MT2) are involved in melatonin's effect on lactose synthesis in mammary cells. Both luzindole (nonselective melatonin membrane receptor antagonist) and 4-P-PDOT (MT2-selective melatonin receptor antagonist) significantly blocked the inhibitory effect of melatonin on mRNA expression of PRLR ($p = 0.028$, $n = 3$, Fig. 5A) but not the mRNA expression of LALBA ($p < 0.001$, $n = 3$, Fig. 5B). Notably, only luzindole but not 4-P-PDOT significantly blocked the inhibitory effect of melatonin on the expression of SLC35B1 ($p < 0.001$, $n = 3$, Fig. 5C). These findings suggested that melatonin receptors were involved in lactose synthesis in mammary epithelial cells.

## DISCUSSION

Several studies have reported that long-term melatonin treatment in cows causes reduced lactose content in milk (*Yao et al., 2020*; *Auldist et al., 2007*; *Molik, Błasiak & Pustkowiak,*

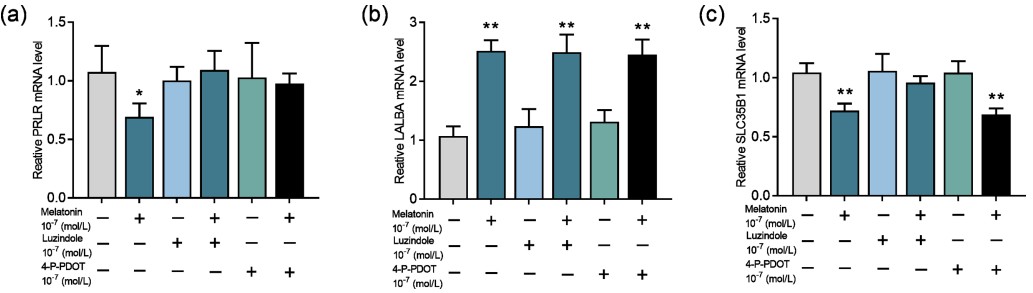

**Figure 5 Effects of melatonin receptors on expression of the lactose synthesis related genes in mammary epithelial cells.** (A–C) Effect of melatonin receptor inhibitors luzindole and 4-P-PDOT on mRNA expression levels of PRLR, LALBA, SLC35B1, re-spectively, data presented as mean ± SEM. ($N = 3$), * $<0.05$, ** $<0.01$ *vs* control.

*2020*; *Yang et al., 2020*). However, the underlining association between melatonin and lactose synthesis in milk has not been investigated. In current study, by daily feeding a fixed dose of melatonin for a long period, we attempt to explore the potential mechanisms of this association . We not only studied the relationship among DHI performance, alterations of serum hormones, and the effects of variable doses of melatonin feeding in cows, but also investigated their potentially signal transduction pathway in mammary epithelial cell culture. To best of our knowledge, this is the first report to show that melatonin's effect on lactose synthesis is mediated by its receptors in mammary epithelial cells and through downregulation of the prolactin receptor gene.

During the experiment, we did not find any abnormalities in the behavior of the animals or in their food intake as previously verified, melatonin has a good safety profile (*Acuna-Castroviejo et al., 2020*). Since melatonin has a very short half-life and the blood samples were collected one day after the final feeding, it is understandable that relatively low doses of melatonin do not result in significantly elevated serum melatonin level because they have been catabolized. And at the same time, high doses of melatonin (for example 10 mg/kg/day) had the elevated serum melatonin levels. Lactose level is a marker of energy balance (*Reist et al., 2002*), we confirmed that melatonin feeding reduced milk lactose content in a dose dependent manner. It seemed that the effective doses ranged from 2.5 to 5 mg/kg/d, while the higher dose of 10 mg/kg/d had lost its effect on milk lactose content. The high dose of melatonin may cause a negative feedback or desensitize the melatonin receptors which are the common features of hoemones, therefore, limits its effect on lactose synthesis . The results are consistent with the previous report, in which low doses of rumen bypass melatonin treatment had a lactose lowering effect while high doses did not (*Yao et al., 2020*). This is not surprising if the effect is the receptor mediated since a fully saturated receptor will blunt the reaction.

From the point of view of energy balance, an altered level of lactose will influence the metabolism of NEFA, another marker of energy balance (*Mantysaari et al., 2019*). In the current study, the reduced milk lactose caused by melatonin feeding did not significantly alter the serum NEFA level. Likewise, glucose is a substrate for lactose synthesis (*Bicjerstaffe,*

*Annison & Linzell, 1974*; *Rigout et al., 2002*; *Sunehag et al., 2002*) and its shortage will result in reduced lactose synthesis. Melatonin feeding also did not alter the serum glucose level, nor did it alter the serum urea nitrogen levels. Therefore, the lowered milk lactose caused by melatonin feeding was not due to an energy imbalance since the fat, glucose, and protein metabolisms were normal after melatonin feeding. Glucose entering mammary epithelial cells requires the help of glucose transporter proteins (*Cant et al., 2002*). The results showed that melatonin did not affect the expression of glucose transporter 1 in mammary epithelial cells. These results indicated that melatonin did not affect either the energy balance of the cow or the substrate availability to synthesize lactose.

In addition, several hormones, including prolactin, progesterone, insulin, and growth hormone, participate in the regulation of lactose synthesis (*Shennan & Peaker, 2000*). Thus, all these hormones were measured in the current study. The results showed that melatonin feeding did not alter the levels of insulin and growth hormone but significantly reduced prolactin production. This was consistent with the previous reports that melatonin treatment reduced prolactin production in cows (*Auldist et al., 2007*) and sheep (*Yang et al., 2020*). Prolactin regulation of lactation in dairy cows. But there was no significant difference in milk production in any of the groups during the experimental period. Similar results on alteration of prolactin without affecting milk production have been reported in other studies (*Smith et al., 1974*). Prolactin deficiency can reduce lactose production in rats and cows due to its effect on inhibiting epithelial cell loss, maintaining tight junctions, and differentiation between mammary epithelial cells (*Flint & Gardner, 1994*; *Ollier, Zhao & Lacasse, 2013*). And also, prolactin can augment the synthesis of lactose by increasing glucose uptake and $\alpha$-lactalbumin availability (*Lacasse et al., 2012*). The results suggested that the decreasing prolactin caused by melatonin feeding might be the reason for the reduced milk lactose. Keeping this in mind, the potential molecular pathway of lactose synthesis was explored in cultured mammary epithelial cells. Usually, the effects of prolactin are mediated by its receptors (PRLR). In humans, prolactin activates PRLR to increase lactose synthesis, and the activation of PRLR then upregulated the mRNA expression of UGP2 and SLC35A2 at the onset of lactation (*Mohammad, Hadsell & Haymond, 2012*). In the current study, we found that the mRNA expressions of the *PRLR* and *SLC35B1* were significantly downregulated with melatonin treatment and coupled with decreased lactose production in mammary epithelial cells. The decreased PRLR expression in goat mammary cells and human breast cancer cells had been reported with melatonin treatment (*Zhang et al., 2019*; *Lemus-Wilson, Kelly & Blask, 1995*).

Finally, we tested whether the effect of melatonin on lactose synthesis was mediated by its receptors. As we know, melatonin receptors MT1 and MT2 were expressed in mammary epithelial cells of dairy goats throughout lactation, and melatonin treatment upregulated the expression of MT1 and MT2 and affected $\beta$-casein expression and cell proliferation in mammary epithelial cells (*Zhang et al., 2019*). Melatonin is able to reduce the phosphorylation level of mTOR *via* the MT1 receptor, therefore affecting the synthesis of milk fat in mammary cells (*Wang et al., 2019*). Our results show that the downregulation of the expression of the galactose transporter SLC35B1 by melatonin was mediated by its MT1. In contrast, the expression of LALBA was upregulated by melatonin treatment.

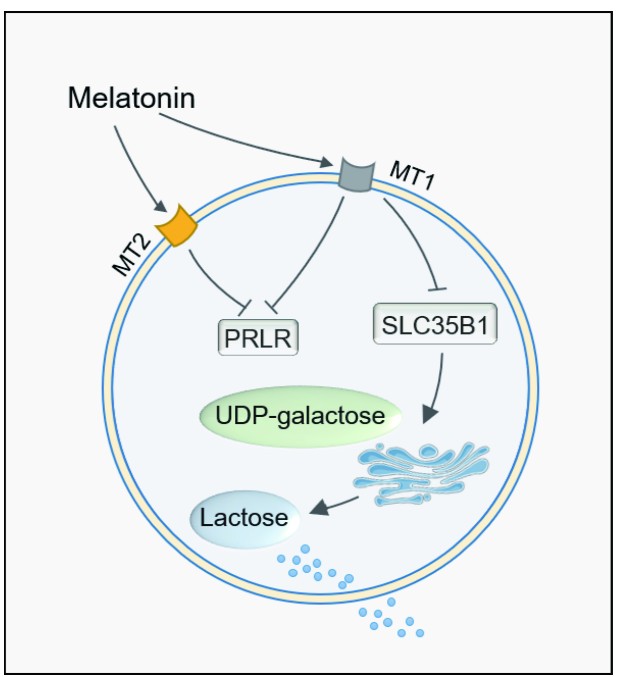

**Figure 6** Path summary diagram of melatonin decrease lactose synthesis action in mammary epithelial cells.

We speculated that the upregulation of LALBA was not the direct effect of melatonin but the physiological compensatory effect of the low level of lactose, which was caused by the insufficient galactose transportation into the Golgi apparatus due to the downregulation of SLC35B1. In lactose synthesis, LALBA expression is more easily regulated by multiple factors such as insulin, while BAG4LT is usually unchanged (*Ribeiro et al., 2023*), which is similar to our results. Therefore, in the current study, we have identified the signal transduction pathway of melatonin in lactose synthesis, *i.e.,* melatonin/MT1/PRLR/SLC35B1/lactose. The observation reveals a novel aspect of melatonin's effect on carbohydrate metabolism in cows or maybe in other mammals, including humans.

## CONCLUSIONS

In conclusion, melatonin long term of feeding reduced the milk lactose content. The transduction pathway was identified to be mediated by the melatonin receptor MT1. MT1 activation downregulates the gene expression of PRLR and SLC35B1. This caused the galactose not being transported to the Golgi apparatus for lactose synthesis and reduced milk lactose content after melatonin feeding. The detailed mechanisms are illustrated in Fig. 6.

## ACKNOWLEDGEMENTS

We thank the staff of Sunlon and Tianquan Dairy Farm for their help with the experiment.

### Funding
This research was funded by the project of Beijing Sunlon Food Group Co., LTD (SNZL202002), the Beijing Innovation Consortium of Livestock Research System (BAIC05-2022), the Future functional food research and development plan (SJ2021002003), the Hebei Province Dairy Cow Industry Innovation Team Project of Modern Agricultural Industry Technology System of (HBCT2018120204), and the Construction subsidy of Hebei Cattle Industry Technology Research Institute (215790557H). The funders had no role in study design, data collection and analysis, decision to publish, or preparation of the manuscript.

### Grant Disclosures
The following grant information was disclosed by the authors:
Beijing Sunlon Food Group Co., LTD: SNZL202002.
Beijing Innovation Consortium of Livestock Research System: BAIC05-2022.
Future functional food research and development plan: SJ2021002003.
Hebei Province Dairy Cow Industry Innovation Team Project of Modern Agricultural Industry Technology System: HBCT2018120204.
Construction subsidy of Hebei Cattle Industry Technology Research Institute: 215790557H.

### Competing Interests
Wenjuan Wei and Changwang He are employed by Beijing Sunlon Livestock Development Co., Ltd. Shujing Li, Wenli Yu, Zengyuan Zhao are employed by Shijiazhuang Tianquan Elite Dairy Co. Ltd. The authors declare there are no competing interests.

### Author Contributions
- Yunjie Liu conceived and designed the experiments, authored or reviewed drafts of the article, and approved the final draft.
- Songyang Yao conceived and designed the experiments, authored or reviewed drafts of the article, and approved the final draft.
- Qinggeng Meng analyzed the data, prepared figures and/or tables, and approved the final draft.
- Xuening Liu analyzed the data, prepared figures and/or tables, and approved the final draft.
- Huigang Han analyzed the data, prepared figures and/or tables, and approved the final draft.
- Chunli Kan analyzed the data, prepared figures and/or tables, and approved the final draft.
- Tiankun Wang performed the experiments, authored or reviewed drafts of the article, and approved the final draft.
- Wenjuan Wei performed the experiments, authored or reviewed drafts of the article, and approved the final draft.

- Shujing Li performed the experiments, authored or reviewed drafts of the article, and approved the final draft.
- Wenli Yu performed the experiments, authored or reviewed drafts of the article, and approved the final draft.
- Zengyuan Zhao performed the experiments, authored or reviewed drafts of the article, and approved the final draft.
- Changwang He performed the experiments, authored or reviewed drafts of the article, and approved the final draft.
- Guoshi Liu conceived and designed the experiments, authored or reviewed drafts of the article, and approved the final draft.

## Animal Ethics

The following information was supplied relating to ethical approvals (*i.e.*, approving body and any reference numbers):

China Agricultural University Laboratory Animal Welfare and Animal Experimental Ethical Inspection Committee provided full approval for this research.

## Data Availability

The raw measurements for Figures 1–5 are available in the Supplementary Files.

## Supplemental Information

Supplemental information for this article can be found online at http://dx.doi.org/10.7717/peerj.15932#supplemental-information.

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
