# Peer review of "A novel signaling transduction pathway of melatonin on lactose synthesis in cows via melatonin receptor 1 (MT1) and prolactin receptor (PRLR)"

_PeerJ, doi:10.7717/peerj.15932_

## Round 0.1 · original submission · Minor Revisions

Please fully respond to the reviewers' comments point by point.

Reviewer 1 ·

Basic reporting

The doses of 2.5 mg/kg/d and 5 mg/kg/d 
melatonin treatment lowed milk lactose content, but not in 10 mg/kg/d 
melatonin treatment. Why is 2.5mg melatonin effective, and 10mg melatonin eliminates that effect?
From what dose of melatonin administration is effective?
It is necessary to examine more smaller doses of melatonin administration.

The doses of 2.5 mg/kg/d and 5 mg/kg/d
melatonin treatment did not increase the blood levels of melatonin, but only the doses of 10 mg/kg/d
melatonin increased serum melatonin levels.
Is the lactose-lowering effect considered to be due to the melatonin (2.5 mg/kg/d and 5 mg/kg/d) effect, even if blood melatonin levels are unchanged?

The melatonin group showed a decrease in blood PRL levels.
Does low levels of PRL affect the decrease of milk lactose content?
Is PRL not involved in lactose production in milk?
Were the amounts of milk secretion unchanged in each group?

Experimental design

.

Validity of the findings

.

Reviewer 2 ·

Basic reporting

From the description in the introduction and discussion, it seems that the effect of melatonin on lactose as well as prolactin has long been discovered. Is it possible to argue that the discovery that melatonin affects galactose transport is the only aspect of originality.

Experimental design

Why choose feeding over injection or implantation when it seems that this method is neither economical nor convenient?

Validity of the findings

(1): line109, it seems that only fig3C significance was counted but not fig3A, please explain why.
(2): The statistical analysis section needs to be described in more detail.
(3): line125, what are the culture conditions of the MAC-T cell referenced? References need to be cited here.

Additional comments

How long does a half-life of melatonin have in the cow blood?

Reviewer 3 ·

Basic reporting

no comment

Experimental design

no comment

Validity of the findings

no comment

Additional comments

Additional comments

This manuscript entitled “A novel signaling transduction pathway of melatonin on lactose synthesis in cows via melatonin receptor 1 (MT1) and prolactin receptor (PRLR)” presented the study on supplementation melatonin to dairy cows would influence the production performance and relevant hormones of cows. Results found that melatonin decreased milk lactose percentage, the levels of prolactin (PRL) and gene expression of SLC35B1.
The MS is qualified to be published in PeerJ after minor revision as follows:
The method of feeding melatonin in the paper needs to be described in more detail.

Line 134-136, The duration of MAC-T cell treated with melatonin and luzindole or 4P-PDOT should be mentioned detailly.

Line 200-203, LALBA mRNA expression changes, but BAG4LT does not change, why?

---

## Round 0.2 · Minor Revisions

The authors have addressed the points in their rebuttal letter quite well, but these issues should be incorporated into the text of the manuscript, and largely they have not.

Please revise the paper further to clarify these specific points in the text. If the reviewers have raised them, I am sure others will be interested in the response.

---

## Round 0.3 · accepted · Accept

The authors have fully responded to the comments of the reviewers and quality of the revised manuscript is significantly improved.